# Structuring Uncertainty for Fine-Grained Sampling in Stochastic Segmentation Networks

**Frank Nussbaum**[*]
Friedrich-Schiller-University Jena
Fürstengraben 1, 07743 Jena, Germany
and
German Aerospace Center (DLR)
Institute of Data Science
Mälzerstraße 3-5, 07745 Jena, Germany
`frank.nussbaum@uni-jena.de`

**Jakob Gawlikowski**[*]
Technical University of Munich
Arcisstraße 21, 80333 Munich, Germany
and
German Aerospace Center (DLR)
Institute of Data Science
Mälzerstraße 3-5, 07745 Jena, Germany
`jakob.gawlikowski@dlr.de`

**Julia Niebling**
German Aerospace Center (DLR)
Institute of Data Science
Mälzerstraße 3-5, 07745 Jena, Germany
`julia.niebling@dlr.de`

## Abstract

In image segmentation, the classic approach of learning a deterministic segmentation neither accounts for noise and ambiguity in the data nor for expert disagreements about the correct segmentation. This has been addressed by architectures that predict heteroscedastic (input-dependent) segmentation uncertainty, which indicates regions of segmentations that should be treated with care. What is missing are structural insights into the uncertainty, which would be desirable for interpretability and systematic adjustments. In the context of state-of-the-art stochastic segmentation networks (SSNs), we solve this issue by dismantling the overall predicted uncertainty into smaller uncertainty components. We obtain them directly from the low-rank Gaussian distribution for the logits in the network head of SSNs, based on a previously unconsidered view of this distribution as a factor model. The rank subsequently encodes a number of latent variables, each of which controls an individual uncertainty component. Hence, we can use the latent variables (called factors) for fine-grained sample control, thereby solving an open problem from previous work. There is one caveat though–factors are only unique up to orthogonal rotations. Factor rotations allow us to structure the uncertainty in a way that endorses simplicity, non-redundancy, and separation among the individual uncertainty components. To make the overall and factor-specific uncertainties at play comprehensible, we introduce flow probabilities that quantify deviations from the mean prediction and can also be used for uncertainty visualization. We show on medical-imaging, earth-observation, and traffic-scene data that rotation criteria based on factor-specific flow probabilities consistently yield the best factors for fine-grained sampling.

---

[*]both authors contributed equally

36th Conference on Neural Information Processing Systems (NeurIPS 2022).

# 1 Introduction

Semantic Segmentation is the computer vision task of assigning a class to each pixel of an image. Examples for popular applications are the segmentation of medical images [1, 2, 39], land-cover classification in earth observation [28, 37, 42], and segmentation of imagery taken by autonomous vehicles [14, 25, 43]. Semantic segmentation tasks are affected by heteroscedastic (input-dependent) aleatoric uncertainty [26, 27] that is also called data uncertainty. Aleatoric uncertainty can emerge in form of label noise in the training data, differing expert opinions about the true segmentation, or ambiguity already contained in the data, for example, caused by technical restrictions like the image resolution [16, 20, 27].

To account for the prevailing aleatoric uncertainty, various probabilistic architectures have been proposed [3, 21, 24, 27, 28, 31]. The work [11] demonstrates that in general, uncertainties predicted by probabilistic segmentation architectures correlate positively with estimation errors, including those obtained from ensemble methods [23, 29] and MC-dropout [15]. Nevertheless, deterministic segmentation architectures [7, 35] are still pre-dominantly used [2, 44]. This may be because for practitioners it is often not clear how to take advantage of the predicted uncertainties. For instance, a limited number of sampled segmentations usually does not represent the predicted uncertainty well. As shown in Figure 1 (top), overview plots for the predicted uncertainty like entropy [11, 27] can be generated. They indicate areas of high uncertainty, where practitioners should take care. However, they do not convey pixel-wise correlations of uncertainty, that is, how changes in the segmentation of one region of an image affect changes in another. Moreover, overview plots cannot explain the overall uncertainty in terms of smaller uncertainty components, which would be desirable towards an interpretation of the uncertainty following the principles of problem decomposition and divide-and-conquer [30].

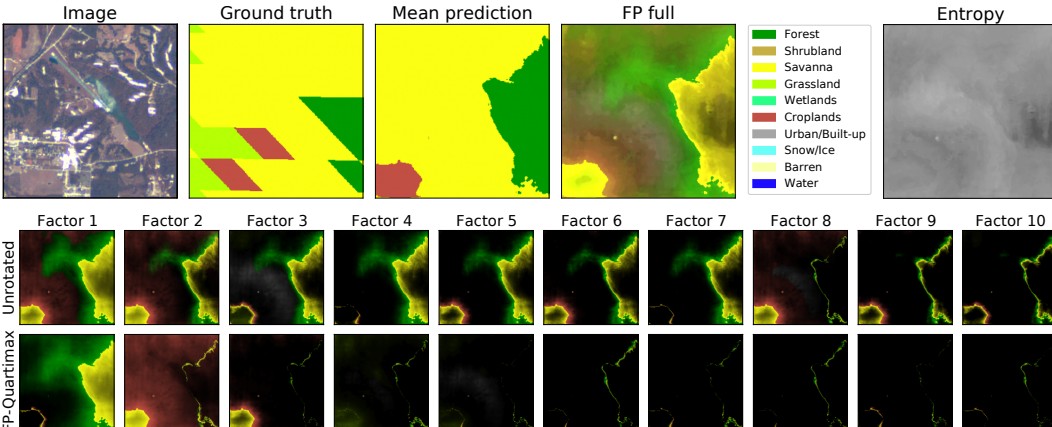

Figure 1: Segmentation uncertainty for an image from the SEN12MS data set [37]. Top row: image, ground truth with coarse labels, mean prediction, overview plot for the predicted uncertainty using full flow probabilities (FP), entropy (bright colors mean high entropy). Bottom rows: Factor-specific flow probabilities for the unrotated factors from the network output (top) and FP-Quartimax factors (bottom). Bright colors mean that the prediction changes with high probability to the class indicated by the color. Flow probability rotations like FP-Quartimax greatly reduce redundancy. Here, they yield only three distinguished factors that encode larger uncertainty components and can be used as the main controls for fine-grained sampling, which was requested in [31].

In the context of the recently introduced *stochastic segmentation networks* (SSNs) [31] that we explain below, the lack of understanding the overall uncertainty in terms of independent components connects to a problem that was also observed by the authors of [31]. They pointed out in their accompanying demo that 'fine-grained sample control' is still missing. Indeed, the identification of independent or at least reasonably distinguished components of uncertainty represents a natural solution to this problem. We seek such components for SSNs with the goal of manipulating them individually for generating and fine-adjusting segmentations.

SSNs model uncertainty via a low-rank multivariate Gaussian distribution on the logits, that is, on the network output before the softmax is applied, see Section 2. Surprisingly, it has not been made use of the fact that this uncertainty model itself offers a straightforward way for distinguishing between uncertainty components. This may be because the low-rank model was originally proposed for reducing the number of parameters. Here, we center our approach around the semantics of the low-rank model as a factor model. Factor models structure the overall uncertainty into individual components of uncertainty, each of which is governed by a single latent variable (called factor). Therefore, we solve the open problem from [31] by using the latent factor variables as control variables for a systematic exploration of the predicted heteroscedastic segmentation uncertainty. For the best result, however, some additional work is necessary: As is known from exploratory factor analysis [13, 40], the latent variables in factor models are only unique up to orthogonal rotations. Hence, they should be rotated for increased interpretability [6, 9, 40], which in our case amounts to generating more useful controls.

Good controls encode uncertainty components that are simple, non-redundant, and separable in the sense that they affect distinguished image regions or classes. To evaluate these aspects, in Section 3 we introduce flow probabilities that quantify deviations from the mean prediction, which also enables uncertainty visualization, see Figure 1. Specifically, we compute *factor-specific* flow probabilities that quantify the impact of the uncertainty components encoded by individual factors. In Section 4, we fuse factor-specific flow probabilities with classic rotation criteria [6, 10] from exploratory factor analysis. In Section 5, we show that these fused criteria generally result in the best possible controls. Note that as a by-product, computing *full* flow probabilities for the overall uncertainty also yields a new type of overview plot that does not aggregate class-specific information about the uncertainty, see Figure 1 (top).

Before we summarize our main contributions, we would like to emphasize that we do not benchmark SSNs as they have already been proven to produce state-of-the-art results w.r.t. various metrics, for instance, generalized energy distance to the ground truth distribution [24, 31]. Hence, it is safe for us to assume that after successful training, SSNs are capable of predicting the aleatoric uncertainty for a given input image reasonably well. Based on that our main contributions are:

(1) control variables for the contributions of individual, factor-specific uncertainty components for fine-grained sampling (given by the latent factor variables),

(2) flow probabilities for quantifying and visualizing overall and factor-specific uncertainties,

(3) rotations based on factor-specific flow probabilities, which structure the uncertainty components and thereby provide simpler, less redundant, and well-separated control variables.

Please find an overview figure of our contributions in Section A of our supplement. Additionally, we made the code for the proposed methods and experiments available under https://github.com/JakobCode/StructuringSSNs.

## 2 Factor modeling in stochastic segmentation networks

Stochastic segmentation networks (SSNs) [31] are characterized by modeling the pixel-wise logits in the network head as a low-rank multivariate Gaussian distribution, that is,

$$p(\boldsymbol{\eta} \mid \boldsymbol{x}) \sim \mathcal{N}\left(\boldsymbol{\mu}(\boldsymbol{x}), \boldsymbol{\Gamma}(\boldsymbol{x})\boldsymbol{\Gamma}(\boldsymbol{x})^{\top} + \boldsymbol{\Psi}(\boldsymbol{x})\right).$$

Here, $\boldsymbol{\eta} \in \mathbb{R}^n$ are the $n = hwc$ logits for an input image $\boldsymbol{x}$ of size $h \times w$ and a classification problem with $c$ classes. The parameters of the Gaussian distribution are the mean $\boldsymbol{\mu}(\boldsymbol{x}) \in \mathbb{R}^n$ and the covariance matrix, which decomposes into a matrix of rank bounded by $r \ll n$ with square root $\boldsymbol{\Gamma}(\boldsymbol{x}) \in \mathbb{R}^{n \times r}$ and a diagonal matrix $\boldsymbol{\Psi}(\boldsymbol{x}) \in \mathbb{R}^{n \times n}$ with positive diagonal elements. The parameters of the Gaussian distribution are the output of a backbone segmentation network with input $\boldsymbol{x}$ [7, 35].

Originally, the low-rank parameterization was solely introduced as a means for reducing the number of parameters [31]. However, the low-rank covariance model has a deeper structural meaning as a factor model [13, 38, 40]. Factor models are characterized by a typically small number of latent variables, called factors, that explain all correlations among a larger number of observed variables. In our case, the joint distribution of the observed logits and the latent factor variables $\boldsymbol{z} \in \mathbb{R}^r$ is given by

$$(\boldsymbol{\eta}, \boldsymbol{z}) \sim \mathcal{N}\left(\begin{pmatrix} \boldsymbol{\mu} \\ \boldsymbol{0} \end{pmatrix}, \begin{pmatrix} \boldsymbol{\Gamma}\boldsymbol{\Gamma}^{\top} + \boldsymbol{\Psi} & \boldsymbol{\Gamma} \\ \boldsymbol{\Gamma}^{\top} & \boldsymbol{I}_r \end{pmatrix}\right),$$

where $\boldsymbol{I}_r$ is the $(r \times r)$ identity matrix, and for brevity, we omit the dependence on the input $\boldsymbol{x}$ in the notation from now on. Here, the interactions of the latent variables with the observed logits are described in the matrix $\boldsymbol{\Gamma}$ of factor loadings: Each column contains the loadings of one latent factor variable on the observed logits, yielding structured uncertainty. The loading characteristic becomes clear in the following sampling procedure for the logits from the factor model:

$$\boldsymbol{\eta} = \boldsymbol{\mu} + \boldsymbol{\Gamma}\boldsymbol{z} + \boldsymbol{\Psi}^{1/2}\boldsymbol{\varepsilon}, \qquad \text{where} \quad \boldsymbol{z} \sim \mathcal{N}(\boldsymbol{0}, \boldsymbol{I}_r), \; \boldsymbol{\varepsilon} \sim \mathcal{N}(\boldsymbol{0}, \boldsymbol{I}_n). \tag{1}$$

This procedure results from sampling from the joint distribution $p(\boldsymbol{\eta}, \boldsymbol{z}) = p(\boldsymbol{z})p(\boldsymbol{\eta} \,|\, \boldsymbol{z})$ as follows: First, the latent variables are sampled according to $\boldsymbol{z} \sim \mathcal{N}(\boldsymbol{0}, \boldsymbol{I}_r)$. Second, the logits are sampled from the conditional distribution $\boldsymbol{\eta} \,|\, \boldsymbol{z} \sim \mathcal{N}(\boldsymbol{\mu} + \boldsymbol{\Gamma}\boldsymbol{z}, \boldsymbol{\Psi})$. Subsequently, only the logits are observed.

Sampling the logits as in Equation (1) provides control over the contributions of the different latent factor variables. This invites for an individual manipulation of the factors, enabling fine-grained sampling. However, as pointed out in the introduction, factors should be rotated beforehand because they are only unique up to orthogonal rotations, see Lemma 1 in the supplement. In particular, orthogonal rotations of the latent factor variables do not change the marginal distribution $p(\boldsymbol{\eta})$ of the logits. Indeed, replacing $\boldsymbol{\Gamma}\boldsymbol{z}$ by $\boldsymbol{\Gamma}\boldsymbol{O}\boldsymbol{z}$ for an orthogonal matrix $\boldsymbol{O} \in \mathbb{R}^{r \times r}$ in Equation (1) yields an equivalent sampling procedure. Therefore, one way to understand orthogonal rotations is that they change the basis of the $r$-dimensional affine space $\{\boldsymbol{\mu} + \boldsymbol{\Gamma}\boldsymbol{z} : \boldsymbol{z} \in \mathbb{R}^r\}$ of the (noiseless) logits, where the basis elements are the columns of the factor loading matrix.

## 3 Flow probabilities

In this section, we develop the notion of flow probabilities as our main tool for the analysis of factor models in SSNs. Flow probabilities can roughly be understood as probabilities of deviations from the mean prediction. We use flow probabilities for uncertainty quantification and visualization.

### 3.1 Factor-specific flow probabilities

Because we need to understand and assess individual factors, it is important to analyze and quantify the uncertainty that is encoded in them. A useful tool for that is to compute factor-wise distributions of class predictions, for which we vary an individual latent factor variable $z$ with associated factor loadings $\boldsymbol{\gamma}$ (a column of $\boldsymbol{\Gamma}$, later we use the notation $\boldsymbol{\Gamma}_{:,j}$ for a specific column of $\boldsymbol{\Gamma}$). We keep the influence of all other latent factor and noise variables fixed to zero. Consequently, for $z \sim \mathcal{N}(0, 1)$ we compute the following expected value:

$$P = P(\boldsymbol{\gamma}) = \int E(\boldsymbol{\mu} + \boldsymbol{\gamma}z)p(z)dz \in [0, 1]^{(wh) \times c}. \tag{2}$$

Here, $E(\boldsymbol{\mu} + \boldsymbol{\gamma}z) \in \{0, 1\}^{(wh) \times c}$ is the matrix whose rows correspond to the pixel-wise one-hot encoded class predictions, which are obtained by reshaping the logits $\boldsymbol{\mu} + \boldsymbol{\gamma}z$ into shape $(wh, c)$ and then applying an $\arg\max$ along the class dimension. Note that the class probabilities $P$ can be understood as a function of the factor loadings $\boldsymbol{\gamma}$ since the mean logits $\boldsymbol{\mu}$ are fixed. We solve Equation (2) analytically. For that, we consider pixels separately since for (flat) spatial index $i \in [wh] = \{1, \ldots, wh\}$, the probabilities $p_{ik}$ from the $i$-th row of $P$ only depend on its associated mean logits and factor loadings. Specifically, with the definition $g_{ik}(z) = \mu_{ik} + \gamma_{ik}z$ for $k \in [c]$, for fixed $z$ the predicted class is $\arg\max_k g_{ik}(z)$. Hence, from Equation (2) we get that

$$p_{ik} = p_{ik}(\boldsymbol{\gamma}) = \int \mathbb{1}[k = \arg\max_{k'} g_{ik'}(z)]p(z)dz, \qquad k \in [c] = \{1, \ldots, c\}, \tag{3}$$

where $\mathbb{1}$ is the indicator function. We solve Equation (3) for binary classification first ($k \in \{1, 2\}$). Assuming that only the logits for the class $k = 2$ are learned, we set $g_{i1}(z) = 0$ for consistency in Equation (3). Then, with $\mu = \mu_{i2}$ and assuming that $\gamma = \gamma_{i2} \neq 0$, the probability $p_{i2}$ evaluates as

$$p_{i2} = \int \mathbb{1}[\mu + \gamma z \geq 0]p(z)dz = \begin{cases} \psi(-\mu/\gamma), & \gamma < 0 \\ 1 - \psi(-\mu/\gamma), & \gamma > 0 \end{cases}.$$

Here, $\psi$ is the cumulative distribution function of a standard normal random variable. For the last equality, observe that $-\mu/\gamma$ is the intersection point of the straight line $g_{i2}(z) = \mu + \gamma z$ with the

$z$-axis $g_{i1}$. If $\gamma = 0$, the arg max is not unique and probabilities can be split. For clarity of the technical exposition, we assume that the arg max is unique in the following.

For general multi-class problems, the probabilities $p_{ik}$ in Equation (3) can be derived from the class-prediction function $z \mapsto \arg\max_{k'} g_{ik'}(z)$. In this function, the class prediction can only change at intersection points $z$ of two non-parallel straight lines $g_{ik}$ and $g_{ik'}$, that is, $z = (\mu_{ik} - \mu_{ik'})/(\gamma_{ik'} - \gamma_{ik})$. Generally, if a class $k$ is predicted for some $z$, then all $z$ values for which the $k$-th class is predicted form a non-empty interval $(\underline{z}_{ik}, \overline{z}_{ik}) \subset \mathbb{R}$. The end points of this interval can either be $-\infty$, an intersection point of $g_{ik}$, or $\infty$. In practice, the intervals $(\underline{z}_{ik}, \overline{z}_{ik})$ can be computed by sorting all intersection points and checking the values of the class-prediction function on the resulting partition of the $z$-axis. If a class $k$ is never predicted, we set $\underline{z}_{ik} = \overline{z}_{ik} = -\infty$. Finally, the class probability is given by $p_{ik} = \psi(\overline{z}_{ik}) - \psi(\underline{z}_{ik})$, where we use the conventions that $\psi(-\infty) = 0$ and $\psi(\infty) = 1$. Observe that the formula for binary problems given above is a special case of the one given for $p_{ik}$ here. Overall, we obtain the following result:

**Proposition 1.** *Define $\overline{Z} = (\overline{z}_{ik})$ and $\underline{Z} = (\underline{z}_{ik})$ with entries $i \in [wh]$ and $k \in [c]$. Then, the distribution of predicted classes under variation of the factor with associated loadings $\gamma$ is given by*

$$P(\gamma) = \psi(\overline{Z}) - \psi(\underline{Z}),$$

*where $\psi$ applies the cumulative distribution function of a standard normal variable element-wise.*

Now, to highlight the difference to the prediction from the mean $\mu$, we compute *factor-specific* flow probabilities as

$$F(\gamma) = P(\gamma) - E(\mu) = \psi(\overline{Z}) - \psi(\underline{Z}) - E(\mu) \in [-1, 1]^{(wh) \times c}.$$

Positive entries in the $k$-th column $F(\gamma)_{:,k}$ indicate that the prediction for the corresponding pixels changes with positive probability from the mean prediction to class $k$. Based on this fact, factor-specific flow probabilities enable visualizations of the impact of individual factors, see Figure 1 (bottom rows). The visualizations are obtained by calculating a mixture of class-specific colors with weights given by the (factor-specific) flow probabilities, see the supplement for details.

As factor-specific flow probabilities represent the real impact of a factor on output segmentations, they will also be a key to quality assessment of the factors, see Section 4. For future reference, we denote by $F(\Gamma) \in [-1, 1]^{(whc) \times r}$ the matrix of all factor-specific flow probabilities that is obtained by concatenating the factor-specific flow probabilities $F(\Gamma_{:,j})$ as columns after flattening, where $\Gamma_{:,j}$ is the $j$-th column of $\Gamma$. Finally, since we use the latent factor variables as control variables for fine-grained sampling, it is helpful to also compute *one-sided* flow probabilities that encode the uncertainty for respectively positive and negative values of the latent factor variable.

**Corollary 1.** *Using the notation from Proposition 1, the one-sided factor-specific flow probabilities for a factor with loadings $\gamma$ compute as*

$$F^+(\gamma) = \int_{[0,\infty)} E(\mu + \gamma z)p(z)dz - E(\mu) = \psi(\max(0, \overline{Z})) - \psi(\max(0, \underline{Z})) - E(\mu),$$

$$F^-(\gamma) = \int_{(-\infty,0]} E(\mu + \gamma z)p(z)dz - E(\mu) = \psi(\min(0, \overline{Z})) - \psi(\min(0, \underline{Z})) - E(\mu).$$

## 3.2 Uncertainty quantification for the full factor model

The idea of computing factor-specific flow probabilities for uncertainty quantification and visualization extends to the full factor model. For that, analogous to Equation (2), we compute the distribution of class predictions. However, this time we take the expected value over the full distribution of the logits given in Equation (1):

$$P^{\text{full}} = \int E(\eta)p(\eta)d\eta = \int E(\mu + \Gamma z + \Psi^{1/2}\varepsilon)p(z)p(\varepsilon)dzd\varepsilon \in [0, 1]^{(wh) \times c}. \tag{4}$$

The change from the mean prediction $E(\mu)$ is then given by the *full* flow probabilities, which we compute as $F^{\text{full}} = P^{\text{full}} - E(\mu) \in [-1, 1]^{(wh) \times c}$. Visualizing full flow probabilities as above by weighted mixtures of class-specific colors yields a new type of overview plot for the uncertainty, see Figure 1 (top row) for an example. Though for our work only a by-product, it has the advantage that

it does not aggregate information about class-specific uncertainties, in contrast to overview plots like entropy (see also Figure 1, top row).

In practice, the integral from Equation (4) is difficult to evaluate. This is because the $\arg\max$ in $E(\boldsymbol{\eta})$ technically amounts to determining a maximum of multivariate linear functions. Hence, we approximate the integral using Monte-Carlo integration with $m$ i.i.d. samples $\boldsymbol{z}^{(1)}, \ldots, \boldsymbol{z}^{(m)} \in \mathbb{R}^r$ drawn from $\mathcal{N}(\boldsymbol{0}, \boldsymbol{I}_r)$ and i.i.d. samples $\boldsymbol{\varepsilon}^{(1)}, \ldots, \boldsymbol{\varepsilon}^{(m)} \in \mathbb{R}^{whc}$ drawn from $\mathcal{N}(\boldsymbol{0}, \boldsymbol{I}_{whc})$. The matrix $P^{\text{full}}$ of class probabilities is thus approximated by

$$P^{\text{full}} \approx \frac{1}{m} \sum_{j=1}^{m} E(\boldsymbol{\mu} + \boldsymbol{\Gamma}\boldsymbol{z}^{(j)} + \boldsymbol{\Psi}^{1/2}\boldsymbol{\varepsilon}^{(j)}) \in [0,1]^{(wh) \times c}.$$

The matrix $F^{\text{full}}$ of flow probabilities can be approximated similarly. In the supplement, we show empirically that the diagonal noise term has little impact on the flow probabilities. Hence, we can focus on the structural uncertainty that is induced by the latent factor variables.

# 4 Factor rotations

As pointed out in Section 2, the latent variables/factors in factor models are only unique up to orthogonal rotations. Therefore, it is common practice in exploratory factor analysis to rotate them in order to maximize their interpretability [6, 22, 38]. The factor model in a SSN represents the predicted uncertainty for a given input image, where the factors themselves encode components of the overall uncertainty. We intend to use them as control variables for fine-grained sampling. From that we derive the following quality criteria:

(1) The number of 'relevant' factors should be small, where relevant factors are characterized by having a 'significant' effect on output segmentations.

(2) Relevant factors should be separable from each other in the sense that they encode distinguished uncertainty components.

(3) Each area in the input image should be affected by only few factors.

Here, the first criterion ensures that the number of impactful control variables is reduced to a necessary minimum, and the second criterion requires that the corresponding uncertainty components are distinct. Together, the first two criteria discourage factor redundancy. The last criterion reflects the general requirement of sparsity and simplicity that is also found among Thurstone's rules [41] for simple structure of a factor loading matrix, which is the primary goal in exploratory factor analysis [13]. However, in our case we rather require a simple structure on the matrix $F(\boldsymbol{\Gamma})$ of factor-specific flow probabilities (see Section 3.1) since they measure the actual impact of the factors on output segmentations. In Section 5, we evaluate different rotation criteria that we present in the following.

First, we consider classic rotation criteria. Here, for a factor loading matrix $\boldsymbol{\Gamma} = (\gamma_{ij}) \in \mathbb{R}^{n \times r}$, Crawson and Ferguson [10] defined the CF family of rotation criteria:

$$q_\kappa(\boldsymbol{\Gamma}) = (1 - \kappa) \sum_{i=1}^{n} \sum_{j=1}^{r} \gamma_{ij}^2 \sum_{l:j \neq l}^{r} \gamma_{il}^2 + \kappa \sum_{j=1}^{r} \sum_{i=1}^{n} \gamma_{ij}^2 \sum_{l:i \neq l}^{n} \gamma_{lj}^2, \qquad \kappa \in [0,1].$$

The CF family is a generalization of the widely used orthomax family [17], where the parameter $\kappa$ controls a trade-off between row complexity (first sum) and column complexity (second sum). We focus on popular choices: $\kappa = 1/n$ yields an equivalent version of the Varimax criterion [22], which is the most used method. Intuitively, it tries to maximize the variance of the squared factor loadings. Next, $\kappa = 0$ yields the Quartimax criterion that minimizes the number of factors needed to explain a variable (in our case segmentation uncertainty of a pixel). Finally, $\kappa = r/(2n)$ yields the Equamax criterion that represents a combination of Varimax and Quartimax.

Classic rotation criteria do not consider the actual impact of factors on predicted segmentations because they only take the factor loadings $\boldsymbol{\Gamma}$ but not the mean $\boldsymbol{\mu}$ into account. Therefore, we incorporate factor-specific flow probabilities into rotation criteria by applying a base rotation criterion $q$ on the flow probabilities instead of the factor loadings. Hence, the objective function to be minimized becomes $\boldsymbol{O} \mapsto q(F(\boldsymbol{\Gamma O}))$ instead of $\boldsymbol{O} \mapsto q(\boldsymbol{\Gamma O})$. We call the new family of rotation criteria the FP family. For instance, FP-Varimax applies the Varimax criterion on the flow probabilities.

# 5 Experiments

The purpose of our experiments is to (1) evaluate rotation criteria based on the quality of rotated factors, (2) demonstrate the merits of fine-grained sample control based on reasonably-rotated factors.

**Data sets and training.** First, we use the LIDC data set [1] in its pre-processed version from [28] that contains 2D slices of 3D thorax scans of size $128 \times 128$ pixels. Each slice respectively has four ground truth segmentations from different experts. Second, we use the multi-spectral Sentinel-2 data from the SEN12MS data set [36] with images of size $244 \times 244$ pixels and coarse labels for semantic segmentation of 10 types of land cover. Third, we use the CamVid data set [5], which contains images of road scenes in resolution $480 \times 360$ and is pixel-wise labeled into 11 different classes. Additional details and statistics about the data sets (including splits) can be found in the supplement, where we also detail all training procedures. We respectively use $r = 10$ in our experiments, which accounts for the varying uncertainty in different images and has also been used in [31]. We would like to emphasize again that we do not benchmark SSNs since they have already shown to be state of the art [24, 31]. For examples of uncertainty predictions, see Figure 1, Figure 2, and the supplement.

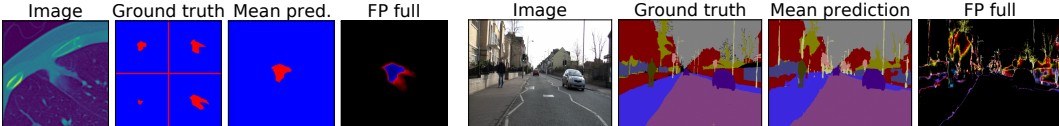

Figure 2: Left: LIDC example including ground truth (split to show annotations of four experts), mean prediction, full flow-probability (FP) overview plot for the uncertainty. Right: CamVid example with uncertainty mostly at class borders. More examples can be found in the supplement.

**Computational aspects.** We used Python 3.7, particularly with the libraries *PyTorch 1.11* [32], *scikit-learn* [33], *NumPy* [18], and *einops* [34]. On a single core of an Intel Xeon Platinum 8260, factor-specific and full flow probabilities can be computed in the sub-second range without significant differences w.r.t. the used rotation, see the supplement for details.

To obtain the optimal rotation matrices for the different rotation criteria, we adapted gradient projection algorithms from [4] to our needs. In our current implementation, optimization for criteria based on flow probabilities can take up to a few minutes, see the supplement for details. In practice, we recommend pre-computing rotations whenever possible.

## 5.1 Evaluation of rotation criteria

We evaluate rotations according to the quality criteria from Section 4, that is, (1) the relevance of individual factors, (2) the separability of the relevant factors, and (3) the sparsity of the factors.

### 5.1.1 Factor relevance

Here, we measure the impact of individual factors on the segmentation. In this section, we use the notation $\tilde{\Gamma}$ to denote a matrix of factor loadings that can be either rotated or unrotated. A simple measure for the impact of the $j$-th factor with loadings $\tilde{\Gamma}_{:,j} \in \mathbb{R}^n$ is given by the $\ell_1$-norm $\|F(\tilde{\Gamma}_{:,j})\|_1$ of the factor-specific flow probabilities. In what follows, we consider relevance curves that show how many factors exceed the overall uncertainty for varying thresholds $\tau \geq 0$. Specifically, we compute

$$n_\tau = |R_\tau|, \quad \text{where } R_\tau = \{j : \|F(\Gamma_{:,j})\|_1 \geq \tau \|F^{\text{full}}(\Gamma)\|_1\},$$

and we measure the overall uncertainty by the $\ell_1$-norm of the full flow probabilities, approximated by 100 Monte-Carlo samples.

**Results and discussion.** The results of averaging $n_\tau$ over the respective test images are shown in Figure 3 (top row). First, classic rotations barely reduce the number of relevant factors compared to the unrotated representation. This is no surprise since they do not take the mean logits into account and only try to simplify the structure of the factor loadings $\tilde{\Gamma}$. Nevertheless, even classic rotations

already seem to decrease redundancy. However, as intended by design, FP rotations reduce the number of relevant factors to a much greater extent. Especially for LIDC and SEN12MS, already small thresholds $\tau$ are sufficient to cut off most factors below the threshold: Figure 3 (top) shows that all FP rotations behave similarly with curves declining sharply for small $\tau$. Consequently, FP rotations tend to produce a huge gap between a small number of relevant factors and the remaining ones, see Figure 1 for a visual example. This is desirable since it allows to focus only on a few relevant and meaningful factors during the exploration of the predicted uncertainty. It may be harder to find such factors if the predicted uncertainty has less inherent structure. CamVid is an example in this regard as uncertainty predictions are often restricted to class borders, which means that they are less spatially correlated. However, even for CamVid, there is structured uncertainty, see Figure 13 (Section D.2.3) in the supplement. Figure 3 (top) shows that also for CamVid, FP rotations significantly reduce the number of relevant factors.

### 5.1.2 Separability of relevant factors

The second quality criterion from Section 4 concerns factor separation. Here, for a separation threshold $\rho \in [0, 1]$, we compute the largest possible fraction of pairwise separated relevant factors:

$$s_\tau(\rho) = n_\tau^{-1} \cdot \max\{|J| : J \subset R_\tau, \cos(F(\tilde{\mathbf{\Gamma}}_{:,j}), F(\tilde{\mathbf{\Gamma}}_{:,j'})) \leq \rho \text{ for all } j \neq j' \in J\} \in [0, 1].$$

If $n_\tau = 0$, we set $s_\tau(\rho) = 0$ for all $\rho$. The separation of two factors is measured by the cosine similarity of their factor-specific flow probabilities, which is always non-negative since corresponding entries cannot have opposing signs. For $s_\tau(\rho)$, a value of one is best since it means that all relevant factors are also separated. For fixed relevance thresholds $\tau$, we also compute the area under the curve $\mathrm{AUC}(s_\tau)$ for the comparison of different rotation criteria.

**Results and discussion.** In Figure 3 (bottom row), we show the separation scores $\mathrm{AUC}(s_\tau)$ for different relevance thresholds $\tau$, respectively averaged over all test images. FP rotations consistently beat classic rotations by a factor of around two in terms of AUC. Classic rotation criteria are still better than the unrotated representations (which form the real baseline). The AUC separation scores drop for thresholds $\tau$ that fail to determine the number of relevant factors sensefully because they are too small or too large. Notably, for classic rotation criteria, the separation scores $\mathrm{AUC}(s_\tau)$ respectively peak at a threshold $\tau$ for which the number of relevant factors nearly coincides with the one from FP rotations, compare the intersection of the curves in Figure 3 (top row). The peak of the separation scores is less pronounced for FP rotations, particularly for LIDC and SEN12MS, where the set of relevant factors is more stable across different thresholds $\tau$. For SEN12MS, the results also distinguish among the FP-rotation criteria, where FP-Quartimax seems to be slightly favored over the

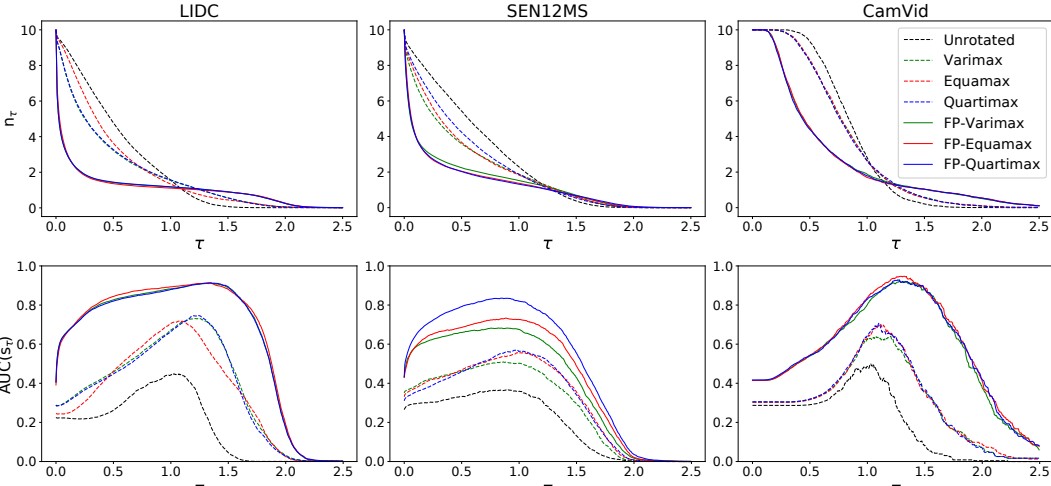

Figure 3: Number of relevant factors $n_\tau$ (top row) and separation via area under the curve $\mathrm{AUC}(s_\tau)$ (bottom row), respectively for different relevance thresholds $\tau$ and averaged over all images from the corresponding test data sets. In the bottom row, high values signify a high degree of separation among the relevant factors. Error bars are deferred to the supplement for visual clarity.

other FP rotations. This may be because Quartimax emphasizes row sparsity the most, which reduces cosine similarities. We investigate row sparsity further in the next section.

### 5.1.3 Factor sparsity

To evaluate to which degree different factors affect the same regions of the input image, we measure row sparsity of the factor-specific flow probabilities $F(\tilde{\mathbf{\Gamma}}) \in [-1, 1]^{(whc) \times r}$. For that, for a (row) vector $\boldsymbol{v} \in \mathbb{R}^r$ let

$$h(\boldsymbol{v}) = \frac{\sqrt{r} - \|\boldsymbol{v}\|_1/\|\boldsymbol{v}\|_2}{\sqrt{r} - 1} \in [0, 1]$$

be the Hoyer measure [19], where values close to one indicate a high degree of sparsity. For us, sparsity only matters in rows with actual uncertainty for the pixel/class, therefore we additionally weigh each row proportional to its $\ell_1$-norm. Hence, as a final measure, we compute the weighted Hoyer measure

$$H(\tilde{\mathbf{\Gamma}}) = \|F(\tilde{\mathbf{\Gamma}})\|_1^{-1} \cdot \sum_{i=1}^{whc} \|F(\tilde{\mathbf{\Gamma}})_{i,:}\|_1 \cdot h(F(\tilde{\mathbf{\Gamma}})_{i,:}).$$

**Results and discussion.** FP rotations generally concentrate the uncertainty for single regions/classes in only few components, see Table 1. This means that FP rotation yield the most disentangled uncertainty components, which also indicates strong separation. For LIDC and SEN12MS, the amount of predicted uncertainty varies greatly across test images, causing high standard deviations. However, in general, large correlating uncertainty components can be found, allowing high row sparsity. This is in contrast to CamVid, where uncertainty is typically predicted for class borders.

Table 1: Mean and standard deviation of row sparsity (100 times weighted Hoyer measure) for different rotations. FP rotations yield the most sparsity. For a more detailed figure, see the supplement.

|          | Unrotated | Varimax | Equamax | Quartimax | FP-Vari  | FP-Equa  | FP-Quarti |
|----------|-----------|---------|---------|-----------|----------|----------|-----------|
| LIDC     | 29.3±14   | 48.2±19 | 37.9±18 | 47.7±19   | 82.8±13  | 81.8±14  | 82.7±13   |
| SEN12MS  | 33.0±17   | 52.7±16 | 50.4±17 | 45.6±19   | 74.1±11  | 74.3±12  | 77.7±13   |
| CamVid   | 26.1± 2   | 30.4± 3 | 30.3± 3 | 30.7± 3   | 50.1± 3  | 50.0± 3  | 49.9± 3   |

### 5.2 Fine-grained sampling

Monteiro et al. [31] already manipulated samples post-hoc by simple linear inter- or extrapolation w.r.t. the mean. However, they noted that additional more fine-grained sample control is necessary for a systematic exploration of the sample space: The interpolation approach lacks a solid foundation in the uncertainty model, and it relies on having a useful sample to start with. The meaningful control variables that we obtain by rotating factors provide all that has been missing. They enable users to systematically explore the sample space by fine-grained sampling: Starting from the mean prediction, they can inspect alternatives, correct possible mistakes, and fine-adjust borders. Particularly, they can

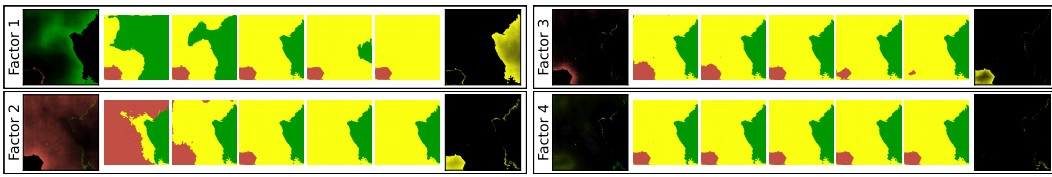

Figure 4: Pseudo-samples obtained by an *individual* manipulation of the four most relevant factors of the FP-Quartimax rotation for the example image from Figure 1. For each factor, five pseudo-samples are respectively obtained by setting the associated factor to $-1, -0.5, 0, 0.5, 1$, while keeping all other factors fixed to zero. The pseudo-sample continuum is intuitively described by one-sided flow probabilities (shown to the left and right of the boxes) derived in Corollary 1. The fourth factor (bottom right) has little impact, which can also be deduced from its nearly zero flow probabilities.

manipulate the contribution of individual uncertainty components by manually setting the values of the corresponding factors. Pseudo-samples obtained in this way are shown in Figure 4.

Alongside this paper, we provide an interface for fine-grained sampling. It allows the selection of a rotation criterion for a given input image (we recommend FP-Quartimax for a start), and control variables can be set conveniently using sliders. In the supplement, we provide some visuals.

## 6 Discussion and conclusion

In this work, we interpreted the uncertainty model of stochastic segmentation networks (SSNs) as a factor model, which provides control variables for fine-grained sampling as requested by the authors of [31]. By (re-)structuring the uncertainty using rotations, we improved the controls and obtained as few as possible, but as many as necessary relevant uncertainty components. Here, it turned out that rotation criteria based on flow probabilities yield the most meaningful controls, where flow probabilities are a new quantification and visualization technique for the uncertainty in SSNs.

Our controls allow to systematically explore the predicted uncertainty and to fine-adjust samples. However, the exploration of the sample space is only useful if the overall predicted uncertainty makes sense. This can be ensured by proper training. Structuring and examining the uncertainty is especially useful if there is a significant amount of aleatoric uncertainty. We note that one limitation caused by our current implementation is that the computation of flow-probability based rotations may take too long for performing it in an interactive scenario. However, there is significant potential for improving the optimization (scheme and parallelization). In any case, rotations should be pre-computed whenever possible.

Overall, we see a broader impact of our approach, which extends beyond the scope of SSNs. For instance, we believe that it could be used for large-scale image classification, where factor models have recently been employed for modeling class correlations [8]. Another promising application is to learn and inspect more structured latent spaces in (variational) autoencoders [12]. Structuring uncertainty as we do may be useful whenever a multivariate Gaussian forms part of a model. Next, flow probabilities can also be used for other probabilistic segmentation architectures that have a mean prediction as a reference point. Notably, flow-probability overview plots for the predicted uncertainty keep class-specific information, in contrast to other overview plots like entropy.

To sum up, it is often easier to understand the whole in terms of smaller parts. In this light, we structured the predicted uncertainty of SSNs into meaningful smaller uncertainty components. Jointly, they enable fine-grained sample control, so for us, the sum of the parts is also greater than the whole.

## Acknowledgements

We thank Prof. Dr.-Ing. Joachim Denzler for helpful comments and Ferdinand Rewicki for checking our work multiple times. We also thank all anonymous reviewers for their insightful feedback.

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
