# OpenReview forum: "Structuring Uncertainty for Fine-Grained Sampling in Stochastic Segmentation Networks"
_NeurIPS.cc/2022/Conference — NeurIPS 2022 Accept_

### Official Review · Reviewer_CT7G · 2022-07-07

**Rating:** 7
**Confidence:** 1
**Soundness:** 3 good
**Presentation:** 3 good
**Contribution:** 4 excellent

**Summary:**

The paper proposes a novel method for structuring uncertainty in the context of stochastic segmentation networks(SSNs). They use low-rank multivariate gaussian distribution to solve SSNs model uncertainty. They also develop a tool for the analysis of factor models in SSNs and apply rotation criteria to provide simple and well-separated control variables. In the experimental part, the proposed method has been outperformed the current state of the art on several datasets.

**Questions:**

How to determine how many factors we should use from Figure 1?

How to determine the ‘significant’ effect where the relevant factors are characterized.


**Ethics Review Area:**

["I don’t know"]

**Limitations:**

one limitation of this work is the heavy computation of the flow-probability rotation.

**Strengths And Weaknesses:**

The idea of this proposed method appears to be a novel combination of stochastic segmentation networks. The authors try to solve the segmentation uncertainty by using smaller latent factor variables based on the recent work called SSNs. The model proposed by the paper is clearly explained and it includes sufficient and detailed experiments in the supplement.


One possible weakness maybe somewhat missing the explanations of the so called ‘significant’ effect on output segmentations; another weakness maybe more detailed description about the rotation criteria. It will be great if the authors could explain more about the interface of fine-grained sampling. In the supplement, the proposed methods seems to have heavy computation cost, I expect them to better address this issue if possible.

---

> ### Author Response · Authors · 2022-08-02
> **Initial response to questions and comments (Reviewer CT7G)**
>
> Thank you for your review and the valuable comments. First, to measure the relevance of a factor, we measure the $\ell_1$-norm of its factor-specific flow probabilities (see Section 5.1.1). To practically determine whether a factor has a ‘significant’ effect, we compare its $\ell_1$-norm relevance score against the overall uncertainty (measured by the $\ell_1$-norm of the full flow probabilities). Intuitively, if the relevance of a factor exceeds a certain fraction of the overall uncertainty, then it can be considered ‘significant’. However, since choosing a particular threshold for the fraction would have been somewhat arbitrary, we instead considered a range of thresholds (see, e.g., the top row of Figure 3).
>
> Nevertheless, it is a good question how one could decide on a number of factors to use/display in practice. Here, although Figure 1 may give a subjective visual cue, for an automatic selection we suggest using the $\ell_1$-norm relevance scores from Section 5.1.1. More specifically, the procedure could be as follows: Descendingly sort all factors according to their relevance score. Then take only as many factors from the most relevant ones until a fraction of, e.g., 95% of the total summed-up relevances is covered by the selected factors. Note that we do not yet use this or a comparable strategy and still display all factors in our fine-grained sampling interface. We do so, for the sake of a demo, to show that after rotation many components are indeed irrelevant. However, for a comprehensive practical application, hiding irrelevant factors that do not have any effect is surely useful.
>
> Concerning the interface you commented that a further explanation of it would be helpful. We agree, so in addition to the video demo that can be found on the starting page of the [submission repository](https://github.com/BlindSubmission2022/StructuringStochasticSegmentation), we now included a more detailed description of the interface in the revised supplement, please see section E.
>
> Finally, we know that the section about the rotation criteria is tight, unfortunately we had to shorten it in the main paper for space reasons. We can recommend the reference [6] from the paper for further reading: In our opinion it contains a nice introduction into the topic and it develops the theory well. Please also note our new overview figure for the whole process in the revised supplement, which may also help.

---

### Official Review · Reviewer_LLek · 2022-07-13

**Rating:** 6
**Confidence:** 4
**Soundness:** 3 good
**Presentation:** 4 excellent
**Contribution:** 2 fair

**Summary:**

The manuscripts reinterprets stochastic segmentation networks (SSN 2020) as a factor model and thus adds latent factors governing the noise components within the single covariance of SSNs. Additionally rotation of the factors with imposed sparsity leads to a parsimonious, and supposedly more interpretable, representation of the factors.  The manuscript provides derivations of the reasoning behind the proposed representation and performs a rigorous empirical comparison of already available rotation approaches. The results in the main manuscript and the supplement, including the video demonstrate that the approach works in providing uncertainty factors that could be individually manipulated.


**Questions:**

- Is there any way to show an intuitive example where the proposed way of displaying uncertainty can be useful, since everything shown is as not helpful as the rial methods of entropy and others?
- Can you report timing information for vanilla SSN vs the proposed modification to give a rough idea how severe is the slowdown?

**Limitations:**

- The value of the proposed approach is not clear from the paper, this limits further impact of the paper since practitioners won't be able to appreciate the need for this method.
- Minor: computational complexity, as noted by the authors.

**Strengths And Weaknesses:**

## Strengths
- A simple and pragmatic approach to extending SSN with an ability to control uncertainty components independently
- A needed take on sparsity and uncertainty to address interpretability of uncertainty representation.

## Weaknesses
- The major weakness, in my view, is that besides meeting the goals of factorization and sparsification of uncertainty, the manuscript is not convincing in that the created tool is interpretable and thus useful. The value of the method for interpretation of the model is not coming through in any of the examples of the manuscript, supplement, and the video.  The most intuitively interpretable examples from the CamVid dataset does not add any information and looks like uncertainty of all classes but, possibly, the cars is simply mixed around all the objects. Other demonstrations are also not helpful and it is unclear how would a user of the model benefit from the new approach in either of the remaining examples. I do not look at the satellite images every day and that may be the reason the flagship example in the main manuscript and the supplied video does not convey much information since the segmentation seems to be very poor (the DICE score would be really low).

---

> ### Author Response · Authors · 2022-08-02
> **Initial response to questions and comments (Reviewer LLek)**
>
> Thank you for your review and the valuable comments.
> First, you mention entropy as a rival method–indeed, it is only useful for getting an overview by displaying the overall predicted aleatoric uncertainty. In contrast, as shown in the new overview figure in the revised supplement, we introduce control variables (factors) for individual uncertainty components that are simple, non-redundant, and well-separated after rotation. This adds the following value beyond mere displaying of the uncertainty (*Question 1*):
> * The independent components facilitate fine-grained sampling, which enables a systematic exploration of the uncertainty and fine-adjustments of segmentations (as opposed to, e.g., reviewing randomly sampled segmentations that are not guaranteed to be representative of the uncertainty). The value of this feature was already recognized in previous work (reference [30] in the paper), where a need for more fine-grained sample control was formulated.
> * The decomposition into independent uncertainty components is also valuable for interpretation: In many cases it is possible to interpret a single rotated component in terms of the distinguished image regions and the usually limited number of classes it affects. For instance, we hope you find the CamVid example from the *new* Figure 13 in the revised supplement intuitive: Its first factor can be interpreted as the structural uncertainty of segmenting a leafless tree in front of a house. Also for the SEN12MS example from the main paper in Figure 4, the first factor can be interpreted as the uncertainty of the class forest (the classification region for forest is either augmented or diminished). Other factors can be interpreted similarly. We believe these types of interpretation to be useful for practitioners because they reveal sub-structures of the uncertainty (correlations) that, e.g., entropy never could. To ease such interpretations, we visualize the effect of factors by factor-specific flow probabilities (see Sec. 3.1 and the new overview plot in our revised supplement). We imagine it to be an interesting extension of our work to automatically derive ‘interpretation’ tags like in the aforementioned examples, which should be possible based on the factor-specific flow probabilities.
>
> For distinction, *full* flow probabilities (Sec. 3.2) also represent an overview plot for the uncertainty and thus are more similar to entropy, although they have the advantage of conveying additional class information by coloring pixels. However, as mentioned in the paper, for us full flow probabilities are more a by-product and not of central importance for our main contributions.
>
> *Question 2*: The time of a forward pass for a Vanilla SSN depends on the backbone segmentation network. As a rough estimate for SEN12MS, a forward pass takes less than a second on our GPUs. On the CPU, time goes up to 16-17 seconds. Our implementation for computing factor-specific flow probabilities currently runs only on the CPU, where it causes an overhead of about 4 seconds. Note that our algorithm for computing factor-specific flow probabilities has complexity $O(r^2 nc)$, $r^2$ for determining the maximum of $r$ straight lines on at most $r^2$ intervals (see the *new* Figure 5 in the revised supplement, $r$ is the number of factors), $n$ is the number of pixels, and $c$ is the number of classes. As mentioned and shown in section D.5 of the supplement, the main computational load lies in the computation of the rotations. For our non-optimized CPU implementation, performing a FP-Quartimax rotation takes about 17 minutes. We are aware that this is a limitation that currently makes pre-computation necessary. We believe that computation times can be reduced greatly by, e.g., improving the optimization procedure, running on GPU, and choosing a more efficient programming language.
>
> Concerning SEN12MS and the flagship example we note that actually only weak (inaccurate) ground truth labels are available (cf. Figure 1). That is why DICE scores are not as high as one would hope for with better labels. The weak labels are a significant source of aleatoric uncertainty, which is why uncertainty analysis is particularly attractive for this type of challenging data. Here, we believe that our proposed fine-grained sampling can be especially helpful in navigating towards a better segmentation. In our demo, we show how practitioners can do so with intuitive sliders for the control variables.

---

> > ### Comment · Reviewer_LLek · 2022-08-08
> > **unconvincing results**
> >
> > Thank you for the explanations. I agree with importance of your intention of having something that usefully quantifies uncertainty of an automatic segmentation. The proposed approach of factoring the uncertainty is also interesting. However, if your examples and demonstrations are the best you can show, I am afraid, your method is not a solution. Based on the results I doubt it will be used in practice.

---

> > > ### Author Response · Authors · 2022-08-08
> > > **Results represent clear improvements**
> > >
> > > Thank you for your comment. In our opinion, our demos and examples serve as proof of concept that users can benefit from the tool we introduced. Concerning your comment–might we ask what you are still missing in our examples and demonstrations? We are willing to further improve based on specific feedback.
> > >
> > > Also please note that the uncertainty quantification is done by the SSN. Our contribution lies in the exploration of the structure of this uncertainty via meaningful user-controlled components. The metrics clearly show that our rotations greatly improve the simplicity, non-redundancy, and separation of the components. Therefore, we believe them to be of value in practice. Of course, the new feature for fine-grained sampling and fine-adjustments of segmentations should be integrated alongside with classic manual techniques for editing segmentations in a full user application. However, in our work we focused on the new technique and not on building a full user application.

---

### Official Review · Reviewer_N33x · 2022-07-25

**Rating:** 7
**Confidence:** 3
**Soundness:** 4 excellent
**Presentation:** 3 good
**Contribution:** 3 good

**Summary:**

This paper tackles the issue of segmentation uncertainty. Using state-of-the-art SSNs, authors view these as factor models. They derive flow probabilities on these factors to visualize and quantify uncertainty associated with them, while looking for a "minimal rotation" of factors through orthogonal rotations.
They show that this technique is suited to derive fine-grained maps for assessing uncertainty in segmentation results, as well as to tweak computed segmentations - which could prove useful for experts using such tools.

**Questions:**

1. Why do you think rotated factors (from FP-Quartimax for instance) seem to be so different across tested datasets? In particular, as you mention at lines 266-268, uncertainty seems to be significant only on class borders for CamVid (cf Fig 11 in the supplemental material), which is the dataset with the highest number of classes. Is your method capable of showing uncertainty for attributed classes not only on borders, and if not, isn't this a major limitation of your model?

2. Are you using loadings or rotated loadings (columns of $\Gamma$ or $\Gamma \cdot O$) in the separation criterion exposed introduced 5.1.2? It is confusing to me that you should use loadings here, when you use rotated loadings line 222.

**Limitations:**

Authors mention computation time of flow-probability based rotation, but I think question 1 could be an important limitation of this work.

**Strengths And Weaknesses:**

I find this piece of work sound and interesting. Flow probabilities (FP), after factors have been rotated in a meaningful way, make for a nice object to intuitively visualize uncertainty and authors show a convincing piece of code allowing users to update segmentations thanks to these FP.

**Significance** The described method could prove very useful if it can indeed help domain experts perform fast segmentation tasks while providing them with intuitive uncertainty measures. However, it is yet unclear to me wether it can help discard entire segmented zones when the number of classes is higher than 2 or 3 (see Question 1).

**Originality** This paper could be considered mildly original as it mostly combines existing results from SSNs, factor analysis and flow probabilities. However, I think the idea to view low-rank models underlying SSNs as factor models sheds a nice perspective on these models, and that this aspect is more important than pure originality.

**Quality** I find this piece of work to be well written and illustrated. I appreciate that authors bundled a repo that I could use out of the box (I tested the notebook and read a great deal of the code). As is, the codebase is hard to grasp and use though, and I think it would greatly benefit from being documented more extensively (docstrings for all methods would be nice) and tested (tests make for a nice way to understand the overall structure of the codebase and usage of each method).
Please cite used packages in the main document (numpy, scipy, sklearn, torch, einops to name a few).

**Clarity** Although this work calls for very visual and easily understood experiments, I found the article a bit difficult to dive in. I think it would benefit from having a figure describing the overall procedure, training steps, and connection between concepts (loadings, factors, latent variables, FPs, rotations and rotation criteria). It could be included in the supp. mat. I think intuitions leading to Proposition 1 would also benefit from being illustrated in a Figure.

---

> ### Author Response · Authors · 2022-08-02
> **Initial response to questions and comments**
>
> Thank you for your review and the valuable comments. First, we agree and created an overview figure for the whole procedure and connections of the concepts, please find it in our revised supplement, Section A. Likewise, we created a Figure for Proposition 1, see Section C.2.
>
> *Question 1*: Our method steps in after SSN training is completed. It has the aim of improving the representation of the predicted uncertainties (which we do by factor rotations). In the process, we take the *overall* predicted uncertainty as it is. Hence, whenever uncertainty is predicted, our method also shows that–particularly, when the uncertainty affects whole regions and not just class borders. This is common for the LIDC and SEN12MS data sets, but also occurs for CamVid, cf. Section D.2 in the revised supplement. However, whether such uncertainty is predicted depends largely on the underlying aleatoric uncertainty in the data, which is different across the tested data sets. This also explains the qualitative differences in rotated factors that you mention. We would like to elaborate this further:
> * For SEN12MS, there is a lot of aleatoric uncertainty for its 10 classes. It is caused by the weak (inaccurate) labels and the ambiguity in the class representations (e.g., classes urban vs. forest when buildings mesh with trees and/or city parks). Hence, there is uncertainty that affects whole regions of images and not just class borders. Especially in the presence of such structured uncertainty, our rotations (cf. Figure 1, paper) can uncover meaningful and non-redundant uncertainty components that allow users to systematically explore the uncertainty for whole image regions. In doing so, users can in principle discard the segmentation of an entire zone by adjusting a component of the factor model that controls this zone (also for data sets like SEN12MS with a larger number of classes). However, such a component only exists if uncertainty is predicted for this zone by the SSN factor model: Our method is designed to explore this predicted uncertainty by fine-grained sampling. Of course, for a comprehensive practical application, a feature that allows a user to manually make *any* changes to a segmentation could be useful.
> * In contrast, for CamVid, the aleatoric uncertainty is indeed mostly at class borders since this data set consists of similar scenes and it has rather accurate labels. As a result, often less overall uncertainty is predicted. However, even for CamVid, there is structured uncertainty. Perhaps we understated this, but to show it we now included an example in Figure 13 (Section D.2.3) in our revised supplement. The first factor in this example can be interpreted as the uncertainty of segmenting a tree without leaves in front of a house (which is a more challenging aspect of the data set).
>
> *Question 2*: The metrics/criteria in the experimental section, among them the separation criterion, are applied on either the unrotated loadings (baseline in Fig. 3) or the respective rotated loadings for the different rotation criteria. We used the symbol $\Gamma$ in all cases in Section 5 because it keeps notation simpler and we also understand $\Gamma$ to be the generic symbol for factor loadings in a factor model (which admittedly slightly overloads notation).
>
> Finally, we also started to address your suggestions concerning the code: Packages are cited, and we wrote more docstrings. Tests will take more time. For now, the pipelines outlined in the readme of the repo (e.g., training, computation and evaluation of rotations) hopefully serve a similar purpose for understanding the code base. To understand how the scripts are related, we created an overview graphic, please find it in the readme of our [repository](https://github.com/BlindSubmission2022/StructuringStochasticSegmentation).

---

> > ### Comment · Reviewer_N33x · 2022-08-08
> > **Some points could still be made clearer in the current work**
> >
> > Thank you for your response. I still think this is an interesting and qualitative piece of work.
> >
> > Here are a few points:
> > * I appreciate that you added these figures to further explain your work! (If you can, try to make the overview figure in the appendix colorblind-friendly - you should mostly change the green-red diverging cmap to something else).
> > * Concerning the notation, I think you should explicitly indicate whether loadings have been rotated or not, otherwise it becomes confusing. Maybe a simple tilde on $\Gamma$ could do the trick? For instance $\tilde{\Gamma}$
> > * I still don't see citations to the libraries you use (torch, sklearn, etc) in your paper: you should cite their original paper or put a link to the repo / doc in footnotes in the main article
> >
> > Moreover, I spent some more time in your code to try and see how much room was left to speed up computation of rotations (I read from your discussion with Reviewer LLek that FP-Quartimax computation takes up to 17 minutes on CPU). Just as a comment, I think you could replace your current svd solver (line 257) with as faster one, and that lines 108-110 of rotator.py might be turned into an ein operation (which this margin is too small to contain).
> > However, I am surprised by line 249 of the same file (the ``while`` loop in ``solve_orthogonal()``), could you elaborate on where this constant comes from?

---

> > > ### Author Response · Authors · 2022-08-09
> > > **Points made clearer**
> > >
> > > Thank you again for reviewing our work so thoroughly and giving such valuable feedback. In a new revision, we addressed all your points:
> > > * We changed the colors in our overview figure to be more colorblind-friendly,
> > > * we introduced the suggested notation in our experimental section and commented on its meaning,
> > > * we added the citations to the main paper (sorry for misunderstanding you in the first place, where we just added specific version links to the readme of the repo).
> > >
> > > Also thank you for your additional suggestions about speeding up the code! Concerning the question about the constant, we added an explanation in the code. Here, we would like to give some additional details. The loop is performing a version of Armijo-Goldstein line search, by default two-way backtracking. Given a search direction (the projected gradient) and an initial step size (learning rate), there are two cases:
> > > * taking a step with the initial step size from the current solver iteration satisfies the Armijo-Goldstein condition (cf. Corollary 2 in reference [1] below, it means that the objective function improves adequately to the step size): In this case, try to double the step size as long as the Armijo-Goldstein condition is still satisfied
> > > * taking a step with the initial step size from the current solver iteration does not satisfy the Armijo-Goldstein condition: In this case, halve the current step size until the condition is satisfied.
> > >
> > > The constant in line 249 is the maximum number of doublings/halvings that our implementation of the line search allows. Typically, line search terminates earlier because one of the other termination criteria mentioned in the two cases above occurs (often less than 5 steps). However, if the initial step size is far off, line search may fail to converge by exceeding the specified maximum number of doublings/halvings (in which case the algorithm issues a warning). In practice, we only observed this for the first iteration when the generic starting value for the step size was too far off. In our experience, this has never harmed convergence since after the first couple of iterations, the step size varies in a smaller range. Hence, the hard-coded constant for the maximum number of line search steps in line 249 mostly barely affects the optimization procedure as it is sufficiently large.
> > >
> > > ----
> > >
> > > [1] Armijo, Larry. "Minimization of functions having Lipschitz continuous first partial derivatives." Pacific Journal of mathematics 16, no. 1 (1966): 1-3.

---

### Meta-Review · Area_Chair_CjAy · 2022-08-26

**Recommendation:** Accept
**Confidence:** Less certain

**Metareview:**

The manuscript interprets stochastic segmentation networks (SSN 2020) as a factor model and thus adds latent factors governing the noise components within the single covariance of SSNs. Additionally, well-chosen rotations of the factors with imposed sparsity lead to a parsimonious, and supposedly more interpretable, representation of the factors. The manuscript provides derivations of the reasoning behind the proposed representation and performs a rigorous empirical comparison of already available rotation approaches. The results in the main manuscript and the supplement, including the video, demonstrate that the approach works in providing uncertainty factors that could be individually manipulated.

According to all reviewers, the paper is well written and the technical work looks solid. The proposed solution is pragamatic. Experiments are sufficient.

The main issue of the paper is that the value of the method for interpretation of the model does not come through the examples of the manuscript, supplement, and the video. It is unclear how would a user of the model benefit from the new approach in either of the examples. The issue is that according to the authors, the new feature for fine-grained sampling and fine-adjustments of segmentations should be integrated alongside with classic manual techniques for editing segmentations in a full user application. However, the present work only focused on the new technique and not on building a full user application. Given this, there remains a doubt wehther the proposed technique is actually useful or not.

Other issues related to presentation clarity and quality of the visual material have been well addressed.

The consensus is that the paper should be accepted at NeurIPS 2022.

**Award:**

No

---

### Decision · Program_Chairs · 2022-09-14

Accept